# Prediction of designer-recombinases for DNA editing with generative deep learning

Lukas Theo Schmitt [1], Maciej Paszkowski-Rogacz [1], Florian Jug [2,3,4] & Frank Buchholz [1] ✉

Site-specific tyrosine-type recombinases are effective tools for genome engineering, with the first engineered variants having demonstrated therapeutic potential. So far, adaptation to new DNA target site selectivity of designer-recombinases has been achieved mostly through iterative cycles of directed molecular evolution. While effective, directed molecular evolution methods are laborious and time consuming. Here we present RecGen (Recombinase Generator), an algorithm for the intelligent generation of designer-recombinases. We gather the sequence information of over one million Cre-like recombinase sequences evolved for 89 different target sites with which we train Conditional Variational Autoencoders for recombinase generation. Experimental validation demonstrates that the algorithm can predict recombinase sequences with activity on novel target-sites, indicating that RecGen is useful to accelerate the development of future designer-recombinases.

Tyrosine-type site-specific recombinases (Y-SSRs) are widely used genome engineering tools. They are capable of exchanging DNA strands between their target DNA sequences, which can facilitate controlled excision, inversion, insertion or exchange of DNA. Because this process is very precise and not reliant on DNA repair mechanisms, it offers seamless manipulation of genomic DNA without side effects (reviewed in Meinke et al.[1]). These features distinguish Y-SSRs from nuclease-based genome engineering methods, such as CRISPR-Cas systems. All nuclease-based genome editing tools rely on the DNA repair pathways of the host cell, which can ultimately lead to unintended editing results (reviewed in Anzalone et al.[2]). However, the advantage of the widely used CRISPR-Cas9 system is the speed and efficiency with which it can be programmed to specifically edit novel target sites. In this regard, Y-SSRs are lagging behind and currently require substantial time and effort to be generated with tailored specificity. This bottleneck represents a considerable hurdle to harness the full potential of designer-recombinases as a versatile genome editing tool.

Current approaches to adapt site-specific recombinases to novel target sequences use directed molecular evolution. One powerful approach to evolve designer Y-SSRs utilizes a plasmid-based bacterial application called substrate-linked directed evolution[3–7] (SLiDE). SLiDE allows libraries of recombinases to be progressively evolved by screening random mutations in conjunction with a selection scheme for activity on new, stepwise altered target sequences. Several recombinases have already been developed this way[4,6–9], demonstrating the applicability of SLiDE. Improvements of the approach have also been reported[7], but the generation of a new designer Y-SSR still requires substantial time and resource investments. The evolution of some of these recombinases took over 150 cycles, an equivalent of 6–12 months of work. Therefore, while SLiDE is effective, it remains expensive in terms of time and labor.

These disadvantages are a direct consequence of the randomized way new generations of recombinases are created when using SLiDE. At the start of a new project, it is not known which sequence modifications will lead to a protein variant that shows recombination activity on the predefined target sequence. There has been some success to introduce improvements based on protein model analysis on designer-recombinases with activity on a defined target site[10]. However, due to the complex nature of the whole enzymatic reaction, which includes DNA binding, DNA bending and catalysis (reviewed in Meinke et al.[1]), it has so far not been possible to rationally design

[1]Medical Systems Biology, Medical Faculty, TU Dresden, 01307 Dresden, Germany. [2]Fondazione Human Technopole, Milano, Italy. [3]Center for Systems Biology Dresden, Dresden, Germany. [4]Max Planck Institute of Molecular Cell Biology and Genetics, Dresden, Germany. ✉e-mail: frank.buchholz@tu-dresden.de

recombinases with activity at a new target sequence. Hence, a method that predicts amino acid changes that will cause recombination activity at a desired target site has the potential to accelerate the generation of novel designer-recombinases.

One way to predict necessary changes within Y-SSRs to achieve altered selectivity could be direct coupling analysis (DCA). DCA models employ probabilistic models to capture co-evolutionary relationships between residues in the protein. Prominent types of algorithms used for DCA are Potts models[11] or Boltzmann machines[12]. They were originally developed for protein residue contact prediction, but they have also been adapted to generate protein sequences[13] and to predict the fitness of a protein[14]. While DCA models are inferred with first- and second-order statistics, they are able to recapitulate third-order statistics[13]. However, there is no obvious way to adapt these models to generate protein sequences conditional for a DNA sequence. Additionally, DCA models can be computationally expensive[12], which could be negatively affected by the added complexity of generating sequences for a DNA target site.

A more recent and more effective approach may be provided by generative deep learning models. These types of algorithms are capable of representing protein sequences as a multivariate distribution whose parameters are learned during training of a deep neural network (reviewed in Wu et al.[15]). Consequently, they can be used for predicting novel proteins by sampling from this distribution. The algorithms that are most commonly used for protein sequence generation are Generative Adversarial Networks (GANs)[16–18] and Variational Autoencoders (VAEs)[19–25]. In addition to these two, there are also deep learning algorithms mostly employed for natural language processing (NLP) that have been adapted for protein sequence generation. In particular, LSTMs[26] and Autoregressive models have been used extensively[27–32]. More recently, transformers have been getting a lot of attention, since they have produced remarkable results in NLP[33] and have also shown great promise in protein sequence generation[34–36]. The multitude of publications on protein sequence generation in recent years clearly indicates the potential of the approach to become a standard for protein development.

Beyond mere generation of new protein sequences, the goal is often to generate variants with improved properties[23,27–30,37]. These improvements usually involve quantifiable aspects, such as thermostability or luminescence. In contrast, the present task requires the generation of recombinase variants with altered properties with respect to a complex categorical condition, their activity on a new DNA target site. To achieve this previously untapped mode of protein sequence generation, we decided to use Conditional Variational Autoencoders (CVAE)[38] to generate models capable of predicting recombinase sequences based on a specific target sequence.

Here we show that the resulting algorithm (RecGen), which we trained with 89 evolved recombinase libraries and their respective target sites, captures the affinities between the recombinase sequences and their respective DNA binding sequences. Through the learned affinities RecGen is capable of constructing recombinase sequences for defined target sites by providing the DNA sequence as a condition. To verify predictions of RecGen, we generated designer recombinases for novel DNA target sites with the algorithm and tested their activity experimentally.

## Results

### Evolved recombinase libraries are highly diverse
An essential part for machine learning is the collection of a large amount of data to train the models. We gathered the training data by sequencing the full gene fragments of evolved recombinase libraries with PacBio HiFi sequencing, a long read sequencing technology that provides highly accurate sequences[39] (Fig. 1a, Supplementary Fig. 1). In total we sequenced 89 recombinase libraries, each evolved for a

different target site (loxA-1 to loxX-6; Supplementary Data 1), covering all four bases on each position of the target sequence half-site at least twice (Fig. 1b). All of these libraries were derived from Cre-based recombinases. For each library, we extracted and translated ~1000 to ~34,000 of the 343 amino acid long recombinase coding sequences, which resulted in a total of over 1.1 million full length reads (Supplementary Fig. 1c).

We first wanted to get an understanding of the data obtained from sequencing the different libraries and assess the quality of the data. We used Pacbio HiFi sequencing to sequence the evolved recombinases, which yielded full length reads with >99.997% accuracy and high reproducibility (Supplementary Fig. 1d, e). Interestingly, the sequenced recombinases showed very high diversity within the libraries, with multiple occurrences of identical clones being rare (Supplementary Figs. 2a, 4b). In all sequenced libraries, mutations found in recombinases were distributed across 206 residue positions (frequency threshold of 10% per library), indicating that 60% of the 343 amino acids were subject to positive selection in at least one evolutionary campaign. Hot spot mutations in the positions 7-15, 30-44, 77, 86-93, 108, 175, 244-249, 258-268, 317-320 had been described previously for individual evolutions of designer-recombinases, providing additional evidence that these regions are important to change target-site selectivity (Fig. 1c; Supplementary Data 2)[1,4,7–9,40,41]. The large amount of sequencing data allowed us to nominate additional mutations that contribute to survival of the selection pressure and potentially to target site selectivity. For instance, we identified positions 5, 23, 57, and 166 as frequently mutated to different residues (Fig. 1c, marked in red), indicating that these residues play important roles in adapting designer-recombinases to a new target site. Hence, specifically targeting these residues in future evolution campaigns could be helpful to generate novel designer-recombinases. Notably, to adapt to activity on a new target site some recombinases acquired up to 75 amino acid substitutions (~22% of the protein sequence) compared to Cre (Fig. 1d, Supplementary Fig. 3), highlighting the adaptability of the protein during evolution. Importantly, we identified residues under positive selection throughout the recombinase coding sequence (Fig. 1c), confirming that virtually the complete sequence contributed to adaptations to new target sites. These results highlight the complex nature of changing target site recognition for this class of enzymes, indicating that simple correlative analyses might not be sufficient to predict new designer-recombinases with custom target site specificity.

Next, we were interested in the composition of the libraries. To get an overview of the distribution of the libraries, we used the dimensionality reduction method t-SNE to visualize the recombinase sequence relationships (Fig. 1d, Supplementary Fig. 2b). We found that many of the evolved libraries form independent clusters, indicating that common changes in the protein led to the ability of the recombinases to recombine the presented target site. However, in some cases multiple clusters were observed for one library, hinting that diverging sets of mutations evolved in the library with activity on the required target site (Fig. 1d, Supplementary Fig. 2c). The complex relationships of the amino acid mutations in regard to the different libraries and therefore the target site selectivity suggested that advanced machine learning might be a way forward to solve the task of recombinase prediction.

### CVAE model architecture for recombinase prediction
The complexity of the data in combination with the requirement of predicting for a target sequence made us disregard the option of using a DCA model. Instead, we turned to deep learning approaches to try and predict recombinases selective for new target sites. There were three requirements we wanted our algorithm to cover. (I) The algorithm needed to be generative, i.e. it should have the ability to predict multiple sequences, so that it can be used in future applications for

library generation and so that it would give us a higher chance of finding functional recombinases. (II) For a deep learning application, the data we have is limited, therefore we were looking for an approach that can be stably trained with the data at hand. (III) Trained models should be capable of receiving the target sequences as input and predict full recombinase sequences as output. Since Generative Adversarial Networks are known to be unstable and difficult to train[42], we decided that requirements (I) and (II) are better fulfilled by Variational Autoencoders (VAEs). More specifically, we found Conditional Variational Autoencoders (CVAE)[38] to be the most suitable method to also fulfill criterion (III), as they are capable of predicting outputs conditioned on a set target sequence.

The CVAE architecture consists of two parts, the encoder and the decoder (Fig. 2a). Our encoder receives the recombinase sequences together with the target sequences in one-hot encoding and learns to represent the given input into a compressed latent space. The latent space is designed to resemble a multivariate normal distribution. Therefore, for each latent space dimension mean ($\mu$) and standard deviation ($\sigma$) are learned for normal distribution sampling (Fig. 2a). In combination with the conditional input of the one-hot encoded target sequence the sampled latent space is delivered to the decoder, which aims to reconstruct the recombinase sequences. During training, the difference between reconstructed and true recombinase sequences are computed via the binary cross entropy loss function. This loss is then backpropagated using the so-called "reparameterization trick"[24]. Additionally, the loss function contains a term that forces all encodings to resemble a multivariate normal distribution (KLD loss, Fig. 2a). Finally, sampling from the controlled latent space in combination with the desired target sequence enables the generation of recombinase sequences through the decoder (Fig. 2b).

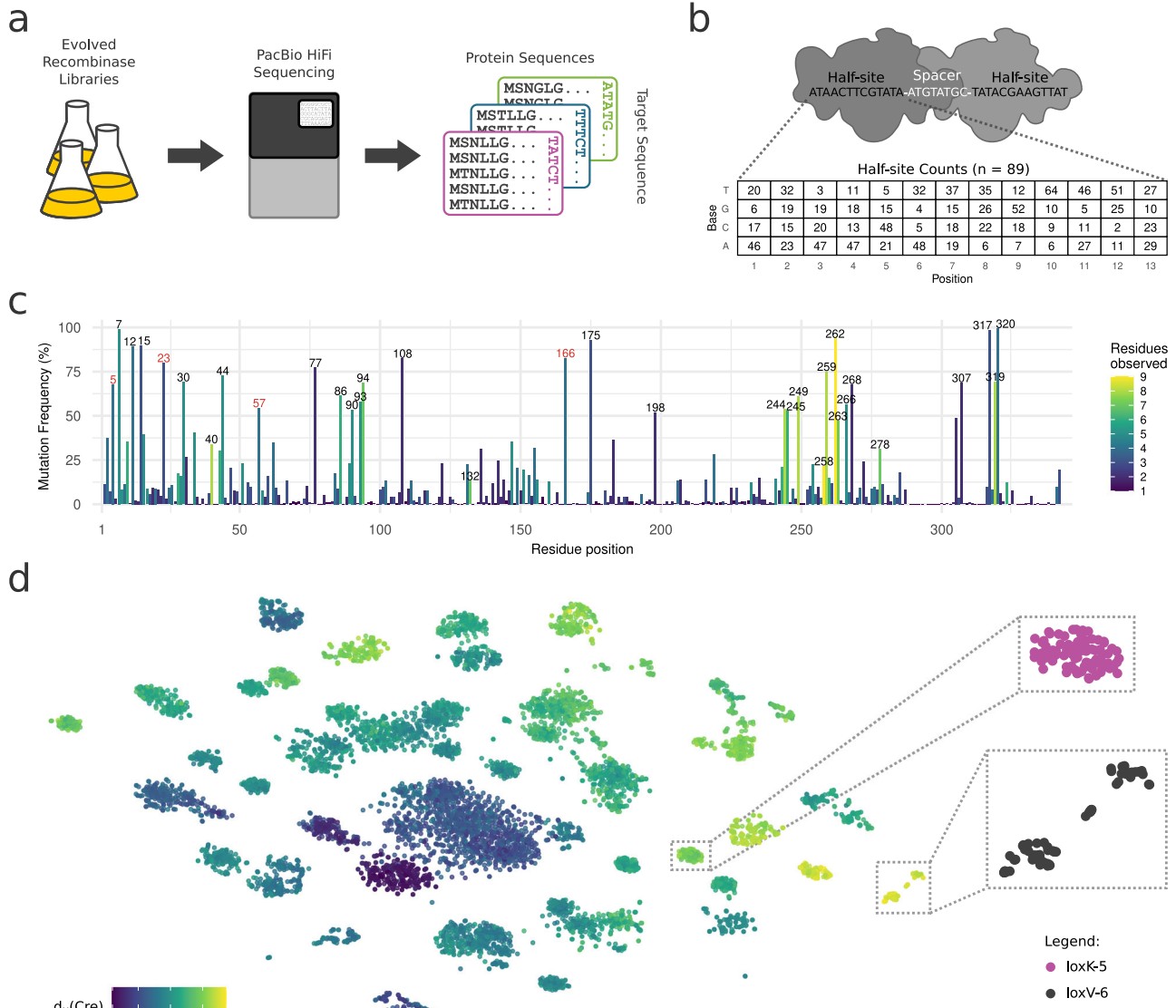

**Fig. 1 | Data acquisition and overview of the recombinase sequence data for training the deep learning approach. a** Illustration describing the data collection. Evolved recombinase libraries were collected and sequenced with the PacBio HiFi method for high accuracy full length reads of the recombinase genes. Gene sequences were translated to protein and stored with the respective target sequences. **b** Illustration of Cre recombinase dimer binding the loxP DNA target sequence (top). All half-site bases covered by the sequenced recombinase libraries are shown on the bottom. The values in the coverage table indicate the number of target-sites with the respective base (rows) on the respective half-site position (columns). **c** Frequency of residues selected for in all sequenced libraries when compared to Cre. Positions mutated in >50 percent of the sequences or with >7 different residues observed are indicated with their number. The number of different amino acids selected for at a particular position is color-coded (Residues observed). Numbers in red had not been highlighted previously. **d** t-SNE dimensionality reduction of 100 random recombinase sequences from all sequenced libraries. Color indicates the amino acid hamming distance ($d_H$(Cre)) of the sequences to Cre. Zoom in on selected libraries on the right. Target site correspondence of zoom ins are indicated. Source data are provided as a Source Data file.

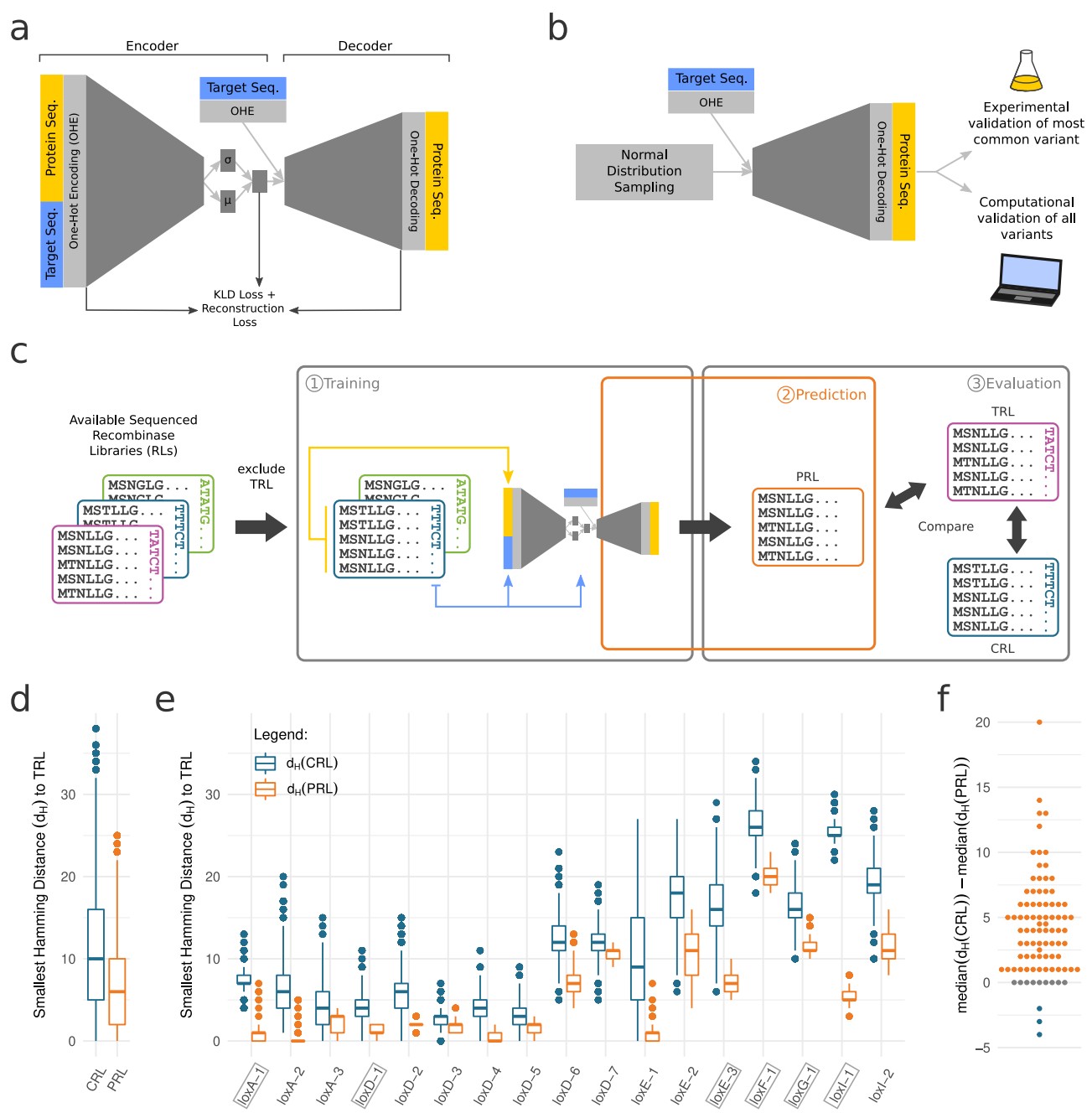

**Fig. 2 | Training and computational validation of RecGen. a** A Conditional Variational Autoencoder (CVAE) is trained with one-hot encoded half-site target sequences and recombinase protein sequences as input. The encoder captures the sequence diversity in a latent space, from which, in combination with the target sequence, the decoder reconstructs the protein sequence. The full model is trained using a reconstruction loss in combination with the Kullback–Leibler divergence (KLD) that ensures the before mentioned diversity of predicted sequences. **b** Trained CVAE decoder generates recombinase protein sequences with the input of a recombinase target sequence and normally distributed random numbers, which diversifies the outcome. The resulting protein sequences for known target sites are computationally validated (explained in part c) and the most commonly predicted sequences for defined targets are validated experimentally. **c** Leave-one-out cross-validation (LOOCV) of CVAE model. (1) CVAE is trained with all available sequenced recombinase libraries excluding the target recombinase library (TRL).

(2) TRL is predicted using CVAE decoder with TRL target sequence and random number (as illustrated in **b**). (3) Predicted recombinase Libraries (PRL) and recombinase libraries with target site closest to TRL (CRL) are compared to TRL. **d**, **e** Boxplots of the smallest hamming distances of each LOOCV predicted sequence to all TRL sequences. Additionally, the CRL smallest hamming distance to TRL is included for comparison. **d** Contains all the CRL and PRL distances joined in one entry each. $n = 89,000$ per condition. **e** Shows the distances of a subset of the libraries. TRL which were used for further experiments are highlighted with boxes. $n = 1000$ per condition. All boxplots are according to standard definition: median for the center line, upper and lower quartiles for the box limits, 1.5x interquartile range for the whiskers, and the points show outliers. **f** Difference of median smallest hamming distances of CRL and PRL. Color codes indicate if the PRL is closer to the TRL than the CRL to the TRL (orange), further away (blue) or the same distance (gray). Source data are provided as a Source Data file.

## Predictions result in high similarity to known targets

To confirm that the chosen CVAE architecture is capable of predicting functional recombinases, we first compared the model output with existing sequencing data. To this end, we decided to use leave-one-out cross-validation (LOOCV), i.e. excluding one target recombinase library (TRL) from the training set and using the resulting model to generate predictions for that one TRL target sequence (Fig. 2c). This procedure was then repeated for each recombinase library.

Assuming that all the sequences in the TRL are capable of recombining the TRL target site, we argued that any predicted recombinase that has a high similarity to any TRL shall be counted as a success. Therefore, to assess the prediction quality, we chose to calculate the hamming distance (i.e. the number of amino acids two sequences differ by) between the predicted recombinase library (PRL) sequences and their closest match in the TRL. In other words, this metric assesses the likelihood that the predicted sequences contain the necessary mutations to enable recombination on the given TRL target site.

Ideally, the model infers the PRL sequences from the mutations it observed from the recombinase libraries in the training dataset with similar target sites. However, there is a chance that it lacks the depth of information to infer a viable sequence, which could lead to 'memorized' sequences from the training data. For this reason, we decided to also measure the distance of the closest recombinase libraries (CRL) in terms of target site distance to the TRL. Hence, if the measured PRL distance is lower than the CRL distance, it indicates that the model was capable of predicting a combination of amino acids with an increased likelihood of working for the desired target site than the amino acid combinations observed in the CRL (i.e. the best otherwise available evolution starting point).

The results obtained confirmed the validity of our proposed approach. More specifically, we calculated the LOOCV PRL hamming distances for all recombinase libraries and found the predicted sequences to be closer to the TRL by a median of four amino acids compared to the CRL (Fig. 2d). In 74 cases the PRL led to median improvements in the distance to the TRL, with only three cases where the median distance of the PRL was worse than in the CRL (Fig. 2e, f; Supplementary Fig. 4a). The highest median improvement was a hamming distance of 20 while in the worst case the distance of the PRL was only four further away from the TRL than the CRL (Fig. 2f). This difference in the distance to the TRL, a known solution to the recombination of the required target site, poses a substantial improvement of the predicted recombinases in comparison to the CRL. This suggests that PRLs are more likely to function at the desired target site compared to CRLs and that PRL sequences need to acquire fewer mutations in directed evolution to recombine efficiently at the desired target site.

## LOOCV predictions generate highly active recombinases

The LOOCV testing indicated that the model was capable of predicting approximations of the TRL sequences, but most of the predictions were still several amino acids different to the actual TRL sequences. We wanted to test some of the LOOCV predicted recombinases to investigate if they are genuinely recombining the respective target sites. In order to assess recombination activity, we used a previously published reporter plasmid[4,7] that can be used in two different assays: a plasmid-based recombination assay (Fig. 3a, b) and a PCR-based recombination assay (Fig. 3c). The plasmid-based assay provides results that can be used to quantify the recombination rate of the recombinases, while the PCR-based assay allows for detection of rare recombination events.

We used the assays to test if the similarity of the predicted sequences to the TRL is an indication of a successful prediction. The predicted sequences for loxA-1 and loxD-1 were very similar to the sequences found in their TRLs (Fig. 2e). This similarity made it likely that these libraries would be functional. Additionally, we investigated

how the predicted sequences compared to the TRLs and the CRLs. We tested the most prominent sequence of both predicted recombinase libraries (PRL single clone; Supplementary Data 3) and found them to have similar recombination efficiencies to the TRLs and the CRLs (Fig. 3d, e). Next, we decided to validate the predictions for loxE-3 and loxI-1, these were more distant to the TRL than the predictions for loxA-1 and loxD-1. However, the $d_H$(PRL) was still much lower than the $d_H$(CRL) (Fig. 2d), suggesting that the predictions might perform better than the CRL if any activity could be observed at all. We found the activity of the most commonly observed prediction (PRL single clone; Supplementary Data 3) for loxE-3 to be higher than the TRL and both CRLs (34% versus 13%, 2% and 7%, respectively, Fig. 3f). On the loxI-1 target site the TRL had an activity of 30%. While both, the PRL and CRL, did not reveal recombination activity in the plasmid assay, recombination could be confirmed with the PCR assay for the PRL, but not the CRL. In summary, for every LOOCV prediction tested, we were able to detect recombination in all predicted recombinases and the recombination efficiency was often comparable or better compared to the CRL, indicating that RecGen is capable of predicting functional and effective recombinases for known target sites.

## Predicted recombinases excise novel target sites

Ultimately, we wanted to be able to predict recombinase sequences with activity on novel target sites, for which no functional recombinase sequences are known. To test whether RecGen can achieve this goal, we decided to generate predictions for 14 artificially constructed target sites for evaluation in a bacterial assay (Supplementary Data 3). We selected target sites from four of our evolved libraries (loxC-7, loxD-7, loxF-1, loxG-1) and changed 3 bp in the half-sites, resulting in loss of activity of most libraries on the modified target sites (Fig. 4). For the changes we focused on the half-site positions 7-12 because these strongly affect the activity of recombinases when they are changed[43]. We also included one novel target site containing modifications in position 3, 5 and 6, because these positions were underrepresented in our training dataset. This way, we constructed 14 target sites for which we generated recombinases with models trained on all recombinase libraries available, thereby setting up controlled test cases to study the performance of RecGen.

For each target site we predicted 10,000 recombinase protein sequences, from which we selected the most prevalent sequence for each target site. Using the recombination assays described earlier we tested the predictions and the CRLs on the respective novel target sites. We investigated putative recombination with the PCR assay and found that most of the CRLs had no activity on the selected novel target sites, indicating that additional sub-sites would be required to start SLiDE for these target sites. In contrast, we detected recombination activity at several target sites of the predicted recombinases. From the 14 novel target sites we found that eight predictions resulted in genuine recombination products which could be confirmed by sequencing (Fig. 4, Supplementary Fig. 5a). In four of the functional predictions no activity could be observed with the CRL, indicating an advantage in using RecGen over the classical method. In addition to testing the predictions on the specified target sites, we also tested their activity on the target site of their CRL. Several of the functional predictions had no activity on their CRL target site (novelC-2, novelD-2, novelF-4; Supplementary Fig. 5b). This is remarkable, because RecGen was developed to predict recombinase activity on a target site, but was not trained to restrict recombination to this site. We concluded that RecGen is capable of predicting recombinases active on defined target sites, making it the first algorithm that can predict functional recombinases for an entirely new target site.

## Discussion

So far, the development of designer-recombinases for novel target sites has been accomplished by applying directed molecular

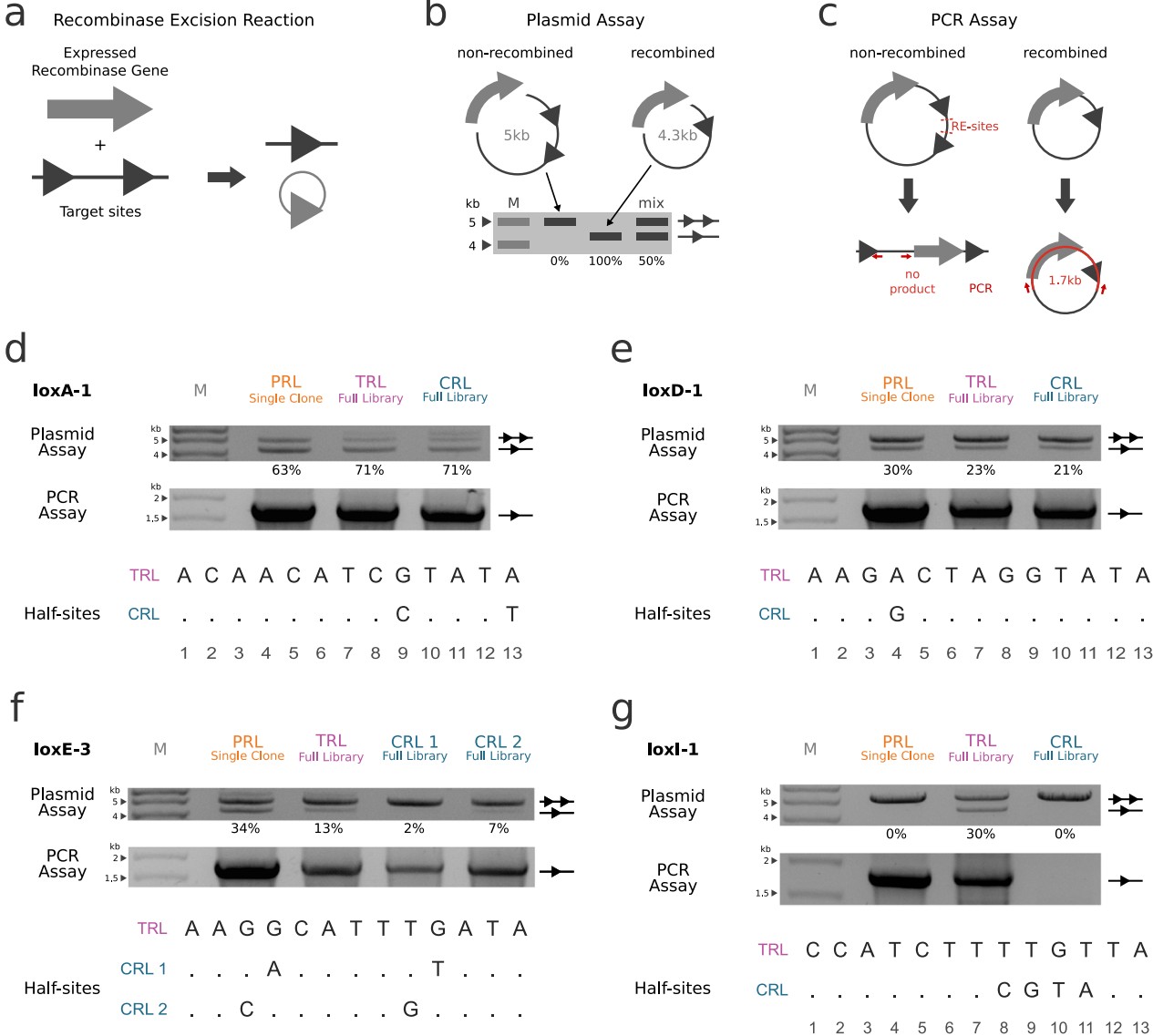

**Fig. 3 | Experimental validation of RecGen leave-one-out cross-validation predictions. a** Illustration of recombinase gene expression leading to excision of DNA with two target sites. The target sites are represented as triangles, two triangles indicate non-recombined DNA, while one triangle indicates recombined DNA. The gray circle with one triangle represents the excised DNA that will be degraded by the cell. **b** Plasmid-based assay for recombination. Reporter plasmids cultured in bacteria cells that express recombinases which excise the DNA between target-sites. The modification leads to a difference of -700 bp in plasmid size, which can be detected via agarose gel electrophoresis. Recombined plasmid fragments are marked with a line and one triangle, non-recombined plasmid fragments are marked with a line and two triangles. "M" indicates the DNA Marker and "mix" is a representation of a mixture of recombined and non-recombined plasmid. Percentages below indicate the amount of recombined plasmid. **c** PCR-based assay for

recombination. Restriction digest with the respective enzymes (RE-sites) leads to linearization of the non-recombined plasmids, while the recombined plasmids stay circular. A PCR with Primers indicated as red arrows results in a PCR product of 1.7 kb from the circular plasmids, while the linearized plasmids produce no products. **d–g** Validation of LOOCV predictions with recombination assays described in **b** and **c**. All samples were tested in the pEVO reporter plasmid with the target site of the respective TRL and cultured at 10 μg/ml L-arabinose. Recombination percentages (below plasmid assay) were calculated from gel band quantifications. The DNA maker is indicated with "M". **d–g** bottom: Alignment of the respective TRL and CRL half-site target sequences. Conserved bases are replaced with a ".". PRL predicted recombinase library, TRL target recombinase library, CRL closest (to TRL) recombinase library. Source data are provided as a Source Data file.

evolution[4–7,9,40,41,44–48], an effective but laborious and time consuming process. A more direct and faster approach could be the intelligent design of Y-SSRs for predefined target sequences. However, Y-SSRs are complex and the understanding of the necessary modifications for a change in target site specificity has been elusive thus far. To tackle this challenge, we gathered over one million recombinase sequences from libraries evolved for 89 different target sites. Consistent with a complex recombination reaction that is hard-wired into the protein sequence, we found that many residues were mutated and positively

selected for in the different libraries. Evolutionary enriched residues are not necessarily important for their target site selectivity. However, the fact that these changes have been found in several independent evolutionary experiments, performed in different laboratories, is strong evidence that they are important for this trait (Supplementary Data 2)[1,4,6,8,9,40,41].

With the extensive sequencing of recombinase libraries we nominate several additional mutational hotspots that seem to influence target site selectivity. Interestingly, a number of mutations

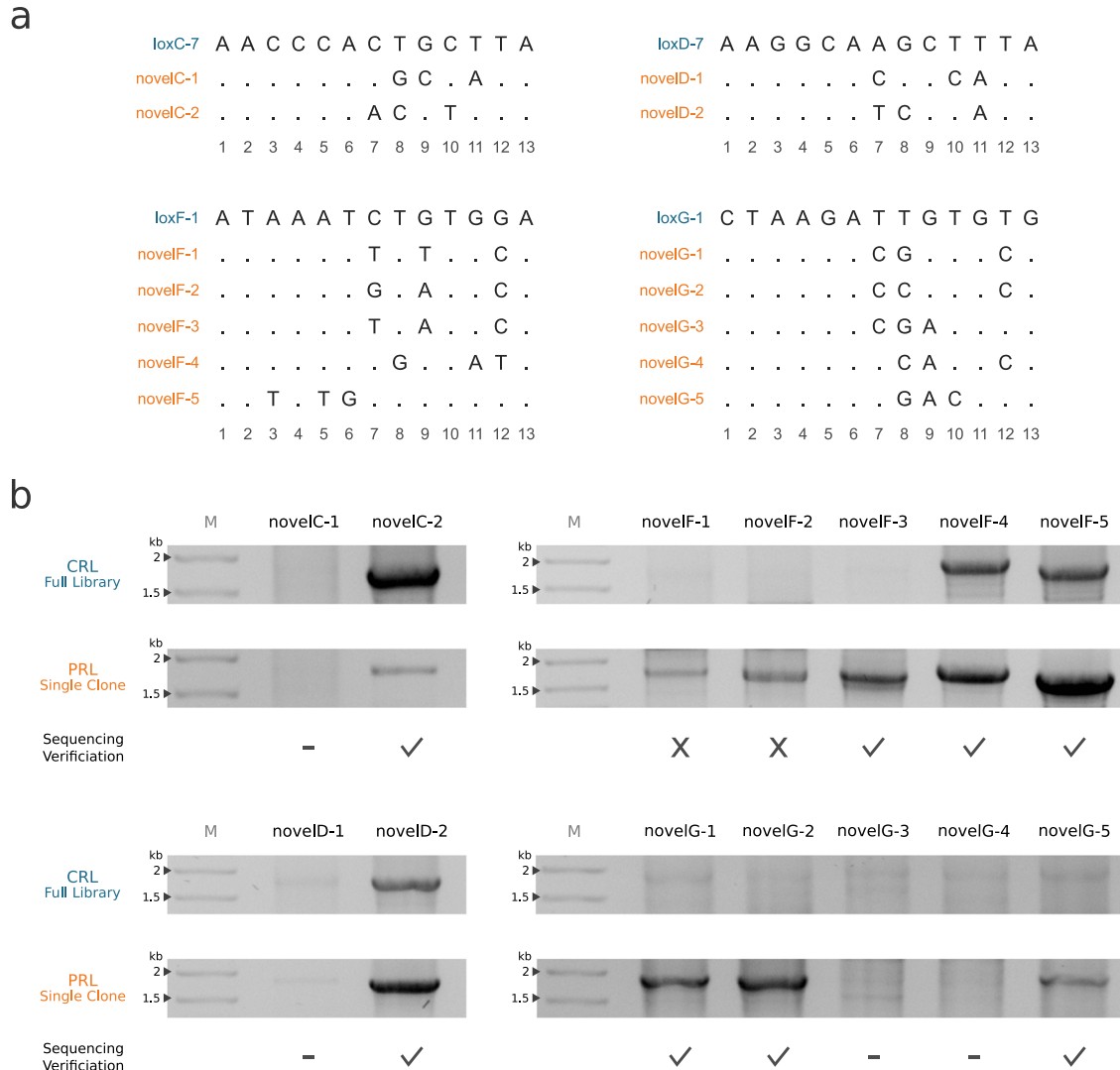

**Fig. 4 | Experimental validation of predicted recombinases for novel target sites. a** Alignment of half-sites from novel target sites with their respective CRL half-sites. Conserved bases are shown with a ".". **b** PCR-based recombination assay of predicted recombinase clones (PRL single clone) and their respective CRL. Check marks indicate that the PCR products contained the edited sequence, while crosses indicate that recombination could not be confirmed by sequencing. Dashes were placed where no sequence verification was performed. PRL: predicted recombinase library, CRL closest recombinase library to novel target site in terms of target site similarity. Source data are provided as a Source Data file.

selected for were found in the N-terminus, a region which is dispensable for recombination in Cre[49,50]. The fact that residues in the N-terminus were clearly under positive selection indicates that the N-terminus fulfills an important role in evolved recombinases[51].

On the library level, we found that some libraries consisted of multiple distinct clusters of sequences, while most others form one uniform cluster. This finding indicates that directed molecular evolution sometimes led to diverging sets of mutations that were capable of recombining the required target site, while other libraries evolved to contain very similar mutations. This trait has been found in other directed molecular evolutions where multiple sets of mutations were dominant until one set took over the whole population[52]. It is therefore possible that for libraries with two clusters additional rounds of SLiDE would be necessary to eventually lead to a single cluster. Overall, sequencing of the evolved recombinases provided us with valuable information that further cemented the complexity of the recombinase target site selectivity.

We decided to approach the non-trivial task of adapting the activity of designer-recombinases to new target sites using a generative deep learning algorithm. With the described neural network we

managed to predict sequences with high similarity to known recombinases specific for the predicted target sites. Through comparing the predictions to the closest recombinase libraries, we could show that these predictions are not merely "memorizations" of sequences with similar target sites. Overall, we found the predicted sequences to be closer to recombinases that are known to work by four amino acids, confirming the validity of our approach.

We did see variance in how closely the predicted sequences resemble recombinases that work and in how much the predictions were improved over the closest recombinase library. We know that the amount of information available for the target site compositions varied and therefore contributed to a difference in the ability of the model to predict the TRL. This could be observed when predictions were performed by leaving out more information related to the TRL (Supplementary Fig. 4c). Additionally, the residues contained in the most closely related libraries could be different even though they fulfill the same purpose. Diverging solutions for the same problem are known to happen in evolution and have also been described in directed evolution[53]. Even though the performance of the models varied in regard to the target site, in over 80% of the tested target sites the

predicted recombinases showed an improvement over the libraries evolved for the most similar target site. This result suggests that the predicted recombinases are generally the better starting point for further directed evolution.

We assumed that the LOOCV predictions closer to the TRL sequences are more likely to be active on the TRL target site. To test this, we validated three LOOCV predictions experimentally. Two predictions that were very similar to TRL sequences were indeed active. However, we found that the two more distant predictions we picked for testing were also able to recombine their target sites. In one of these cases, it turned out that the prediction was even more effective than the library that was evolved for this task. We suspect that part of the sequence difference between the PRL and TRL is made up from rare mutations that the model omitted. Many of these rare mutations have a neglectable effect on protein function and are therefore neither enriched nor removed. Because these mutations are so rare their influence on the training loss is very low, which is why RecGen was not optimized to learn these mutations. This led to the predictions being less variable (Supplementary Fig. 4b), but the important residues were most likely maintained. All in all, the tested LOOCV predictions of novel target sites were all functional, an indication that the predictions from this model are useful.

Ultimately, we wanted to know if the model is capable of fulfilling its intended purpose, the prediction of recombinases for novel target sites. We could confirm that 8 out of 14 predictions were active on the respective target sites. Importantly, most CRLs, which were the libraries we would usually start an evolution from, showed no activity. This means that if we wanted to get a library with activity on the defined target site, it would be necessary to first evolve a library that works on a subsite, so that we could then move on to the target site of interest. Using our approach, we can now start an evolution directly on the intended target site. In recent work[6], evolving recombinases to adapt to a target site with one or two bases changed took 12 and more cycles. Because each cycle is a full working day, each target site switch takes >2 weeks of actual working time. Therefore, our predicted recombinases that have three bases changed in the target site would save at least 2 weeks of work.

The predicted recombinases were also tested for activity on the CRL target site, to probe for their specificity. Since the CRL is the library with the most similar target site in the training dataset, it is reasonable to assume that the model would implement protein sequence features from the CRL. This could result in the prediction being active on the CRL. Indeed, we found that some of the PRL are active on their CRL target site, while several successfully predicted recombinases did not recombine the CRL target site. The current version of RecGen only addresses activity on a specified target site. Hence, further directed evolution, or rational design on these recombinases would have to be performed to make these recombinases highly active and specific[4–6,8,10,54].

One has to note that we only tested a single sequence of the PRLs, while we tested a full library for the CRL. The probability of finding a functional recombinase is much higher in a library of >100,000 variants than when testing only a single clone. Therefore, testing multiple predicted recombinases would probably yield an even higher success rate than just testing one sequence. Overall, the results we obtained here are encouraging and indicate that RecGen will be beneficial in cutting down the time and effort spent for the development of new designer-recombinases.

RecGen as it stands is already a valuable resource to shorten the development time of new designer-recombinases, but future iterations will likely further it's capabilities to reliably predict Y-SSRs. Changing the encoding to amino acid properties or a learned representation for protein sequences[27,55,56] would give RecGen additional information that could help to predict more reliably. Another way to adapt the network would be to change the fully connected neural network layers to convolutional or recurrent network layers, which could further improve the performance[22,23]. Recent publications demonstrated impressive results with transformer models[34–36], indicating that this kind of model could also be useful for recombinase sequence data. Advances in protein structure prediction and generation could help filter the model predictions for proteins that are functionally viable. For example Ferruz et al.[34], validated their predicted protein sequences with AlphaFold2[57] through structure probability scores and MD simulations. While improvements to the algorithm are important, the data used to train the model is likely to be even more critical.

The amount of data we gathered is considerable in terms of laboratory work performed, but the target sequence space where successful predictions can be done is still limited. So far, we could only confirm predictions for target sites that differ by three bases in the half-site to known target sites. This is why the collection of more data will be key in making RecGen more useful for future designer-recombinase development. This means that we need to collect sequences from recombinase libraries evolved for additional target sites. Ideally targeted evolutions specifically for the purpose of extending the available sequence space should be performed. In addition, quantitative data about individual recombinase sequences within the library should be helpful to improve RecGen. This information would be important for the model to understand which mutations are more relevant for recombination activity on the defined target site. Because directed evolution for new functionality could cause recombinases to become more "promiscuous", i.e. less specific[4,58], it would also be relevant to screen for their activity on off-targets. This would help the model to learn how "flexible" the proteins are in their target site selectivity. Generally, the more data available for the model, the better the possible outcome.

In conclusion, we show that RecGen is capable of predicting recombinases for defined target sites. To the best of our knowledge, the successful prediction of proteins selective for a defined DNA target site has not been accomplished before and is therefore an important achievement. Furthermore, we believe that RecGen will be a valuable tool to kick-start the evolution of future designer-recombinases, making the process faster and less laborious.

## Methods

### Deep Sequencing of evolved recombinase libraries

In order to deep sequence the recombinase libraries, 500 ng of pEVO plasmids (described in[3,4,7]) from the evolved recombinase libraries were digested with the restriction enzymes NdeI and AvrII (NEB catalog number R0111S and R0174S), which will only cut in non-recombined plasmids. The digested DNA was then desalted with a MF-Membrane and cultivated in *E. coli* XL-1 Blue (Agilent catalog number 200158). After growing the cells for 14–16 h in 100 ml LB in the presence of chloramphenicol (25 mg/ml), plasmid DNA was extracted using the GeneJet Plasmid Miniprep Kit (ThermoFisher catalog number K0502). The extracted DNA only contains recombined plasmids with functional recombinase genes. 5 ug of the extracted pEVO plasmid were then digested with the restriction enzymes BsrGI-HF and XbaI (NEB catalog number R3575S and R0145S) and the 1041 bp fragment harboring the recombinase gene was then enriched with custom SPRI-beads (described in dx.doi.org/10.17504/protocols.io.n7hdhj6), by binding the pEVO backbone to the beads two times followed by an Ampure XP (Beckman Coulter catalog number A63881) cleanup of the supernatant, a more detailed description can be found in the supplementary methods. The DNA was quantified with a Qubit dsDNA HS Assay Kit on a Qubit 2.0 Fluorometer (ThermoFisher catalog number Q33230) and sent to the DRESDEN-concept Genome Center, where it was sequenced with a Sequel System 6.0 using the Pacbio HiFi method.

Circular consensus sequence data was generated with PacBio's ccs v3.4.1. Using ccs's filtering criteria, only sequences of length 1034-

1200 bp with a minimum predicted accuracy of 99.997% (Phred Score of ~25) were kept. After converting the data to FASTA format using SAMtools 1.11 the recombinase gene sequences were aligned and translated to amino acid using the protein2dna:bestfit alignment model of exonerate v2.3.0. Further filtering was done with GNU grep v3.7, GNU sed v4.8 and GNU awk v5.1.1 to ensure that the genes are 1032 bp long, start with a methionine, and do not contain any stop codons within the sequence.

### Recombinase sequence data analysis

Recombinase protein and target site sequences were processed using R version 4.1.1 with tidyverse packages v1.3.1[59,60]. t-SNE dimensionality reduction was calculated with the Rtsne package v0.1.5 from hamming distances of the sequences acquired with the stringdist package v0.9.8[61,62].

### CVAE model training and prediction

Neural networks were trained using pytorch v1.10.1 on Python 3.9.6[63]. Data pre- and post-processing was done with numpy v1.22.1 and pandas v1.4.0[64,65]. To ensure equal distribution of the training data we used 1000 randomly chosen recombinase sequences per library (Supplementary Fig. 2a) plus the corresponding target half-site, which were one-hot encoded for training of the models.

One-hot encoding was generated by forming a binary matrix where the rows correspond to the recombinase sequences and the columns represent all possible combinations of amino acids and positions in the sequence. The sequence was then represented with ones in the places that correspond to the combinations found in the sequence and zeros for all other combinations. The target sequence and the recombinase sequence were concatenated and the possible letters allowed contained the naturally occuring 20 amino acids and one for the stop codon. The target sequence part was encoded in the same way, since the letters for the DNA are contained in the 20 amino acids.

The training dataset corresponded to 89 target sites from 23 evolution projects. CVAE architecture was constructed according to Sohn et al.[38]. The models were constructed with two fully connected layers in the encoder and decoder with 64 and 32 neurons (reversed in the decoder), while the latent space was defined with two neurons. Training was performed in 40 epochs with a batch size of 128, a dropout rate of 0.1, and a learning rate of 0.0001. The reconstruction loss was calculated with binary cross entropy and the KLD is multiplied with a factor of 0.1 to 2, linearly increasing with each epoch[66].

Predictions with the trained models were done using the decoder part of the model by providing the one-hot encoded half-site target sequence with two random numbers generated from a normal distribution (Fig. 2b). The output of the model is then decoded from one-hot to amino acid. All predicted recombinase sequences can be found in Supplementary Data 3.

### LOOCV testing

Leave-one-out cross-validation was performed as illustrated in Fig. 2c. First the training set is reduced by removing the Target Recombinase Library (TRL). Model training and prediction with the reduced training dataset were then done as described above. We found that training with the same parameters results in slightly different outcomes. To mitigate the output variance we decided to use multiple models for prediction, because of the computational cost we trained 10 models per TRL for prediction. A total of 1000 recombinase sequences were generated and their hamming distances to the TRL sequences was calculated with only the smallest distance recorded. Additionally, the same process is done with the CRL sequences in place of the PRL sequences. The recorded distances were then processed with the tidyverse R packages v1.3.1 and plotted with ggplot2 v3.3.3 [67].

The smallest hamming distance is defined in Eq. (1).

$$d_H(x_i) = \min\left(\Delta\left(x_i, \{y_1, \ldots, y_j\}\right)\right) \qquad (1)$$

Where $x$ is the PRL or CRL sequences and $y$ is the TRL sequences. $\Delta$ indicates the hamming distance function.

### Recombinase activity assays

pEVO vectors with the respective target sites and recombinase genes were cultivated in *E. coli* XL-1 Blue (Agilent catalog number 200158) for 14–16 h in 10 ml LB in the presence of chloramphenicol (25 mg/ml) and L-arabinose (10 μg/ml or 200 μg/ml). Plasmid DNA was extracted using the GeneJet Plasmid Miniprep Kit (ThermoFisher catalog number K0502). For the plasmid-based recombination assay 500 ng plasmid DNA was digested with XbaI and BsrgI-HF (NEB catalog number R3575S and R0145S) followed by agarose gel electrophoresis. The resulting bands indicate the fraction of recombined plasmid at ~4.2 kb and non-recombined plasmid at ~5 kb. The amount of recombined and non-recombined plasmid was measured from the band intensities using Fiji for image processing. The rate of recombination was calculated by dividing the amount of the recombined plasmid with the amount of both plasmids. A higher percentage of recombined plasmids indicates that the recombinase library or variant is more active on the defined target site. For the PCR-based recombination assay 500 ng of the plasmid DNA was digested with NdeI and AvrII (NEB catalog number R0111S and R0174S), which linearizes all non-recombined plasmids. The digested product is then subjected to PCR with the primers EVO-F and EVO-R (CGGCGTCACACTTTGCTATG, AAGGGAATAAGGGCGA-CACG), which will amplify if the product is circular, but will fail if the product is linearized. The expected product of ~1.7 kb is visualized with agarose gel electrophoresis.

### Validation of LOOCV predictions

The pEVO evolution and reporter vector for loxA-1, loxD-1, loxE-3, as well as the respective TRL and CRL genes were obtained from Sarkar et al., Karpinski et al. and Lansing et al.[7–9]. (Supplementary Data 1). The pEVO vector for loxI-1 and the TRL genes were produced for unpublished work by Martin Schneider in the same manner as the above-mentioned vectors. The loxI-1 CRL genes (loxB-1) were obtained from Sarkar et al.[9]. The PRL clones were ordered codon optimized for *E. coli* from Twist Bioscience. All recombinase genes were excised by XbaI and BsrGI digest followed by gel excision (Isolate II PCR and Gel Kit from Bioline, catalog number BIO-52058) for cloning into the respective pEVO vector. The cloned vectors were tested for recombination activity with the assays described above using a concentration of 10 μg/ml L-arabinose.

### Prediction for novel target sites and validation

As mentioned in the results section, novel target sites were generated by randomly mutating three bases of the half-site target sequence at positions 7–12 and selected for a distance to other known target sites of four or more. Additionally, we also produced one target site (novelF-5) with base changes in the positions 3,5, and 6, where the positions were changed to least frequently occurring bases in the training dataset (Fig. 1b). pEVOs with these 15 generated target sites were then constructed as described previously[7]. In short, primers containing the target site and an overlap with the pEVO plasmid were used to produce a PCR product from a pEVO plasmid. This fragment was then cloned into a BglII (NEB catalog number R0144S) digested pEVO plasmid using the Cold Fusion Cloning kit (System Biosciences catalog number MC010B-1). The CRL, which is also the library where the target sites were modified to make the novel target site, was tested for recombination activity (200 μg/ml L-arabinose, described above) on the novel target site pEVO. From the 15 target sites we found 10 to have no activity on their respective CRLs, which were

then selected for testing predicted recombinase sequences. Additionally to those, a further four target sites were generated in the same manner, but without testing activity on their CRLs (novelC-2, novelD-2, novelF-4, novelF-5) and selected for prediction testing. For these 14 novel target sites recombinase sequences were predicted with 100 models trained on the full training data as described above. Each model was tasked to predict 100 recombinase sequences, which resulted in a total of 10,000 recombinase sequences per novel target site. The most prominent predicted recombinase sequences per target sites were then ordered and cloned as described above. Recombinase activity assays were also performed as described above with 200 µg/ml L-arabinose. Sequence verification was performed from gel excision products of the 1.7 kb fragments (Isolate II PCR and Gel Kit from Bioline, catalog number BIO-52058) with the Microsynth sanger sequencing services.

## Statistics and reproducibility

Statistical analysis was performed using R version 4.1.1. All boxplots are according to standard definition: median for the center line, upper and lower quartiles for the box limits, 1.5x interquartile range for the whiskers, and the points show outliers. Relevant details of the statistical test are provided in the figure legends. Representative gel pictures showing recombination activity were not replicated for recombination plasmid assays or recombination PCR assays. No statistical method was used to predetermine sample sizes. No data were excluded from the analysis. The experiments were not randomized.

## Reporting summary

Further information on research design is available in the Nature Portfolio Reporting Summary linked to this article.

## Data availability

The sequence data generated in this study have been deposited in the European Nucleotide Archive under accession code PRJEB57361. Source data are provided with this paper.

## Code availability

The code for RecGen can be found at https://github.com/ltschmitt/RecGen [68].

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

## Acknowledgements

We would like to thank Janet Karpinski, Felix Lansing, Duran Sürün, Teresa Rojo Romanos and Martin Schneider for providing evolved designer-recombinase libraries. Further, we would also like to thank Felix Lansing and Duran Sürün for their suggestions regarding the first draft of the manuscript. Thanks also goes to Joshua Shrouder for proofreading. Furthermore, we thank the DRESDEN-concept Genome Center at the CMCB from TU Dresden for performing the sequencing of the recombinase libraries.This work was supported, in part, by the European Union (ERC 742133—F.B., H2020 UPGRADE 825825—F.B.) and the BMBF GO-Bio (031B0633—F.B.).

## Author contributions

L.T.S.: conceptualization, methodology, software, validation, formal analysis, investigation, data curation, writing—original draft preparation, writing—review and editing, visualization. M.P.-R.: conceptualization, writing—review and editing. F.J.: fonceptualization, writing—original draft preparation, writing—review and editing. F.B.: conceptualization, writing—review and editing, supervision, project administration, funding acquisition.

## Funding

## Competing interests

The Authors F.B. and M.P.-R. are founding members and shareholders of RecTech GmbH that commercializes recombinase technologies. The remaining authors declare no competing interests.
