## [Peer Review File · Nature Communications]

Reviewers' Comments:

Reviewer #1:

Remarks to the Author:

The authors describe the development and preliminary testing of a deep learning approach to predict amino acid sequences of tyrosine recombinases that will have catalytic activity on novel target sites, based on previously existing libraries. Taking their approach from in silico predictions to in vitro analyses of a small set of predicted protein sequences, they observed recombination of novel target sequences in 4 of 10 of the enzyme/target pairs tested. This approach is substantially faster and more accurate than the alternative method of using the closest sequence as a starting point for evolution. Because tyrosine recombinases yield seamless genomic outcomes that don't rely on the stochastic outcomes of cellular repair machinery, they offer a means to achieve more predictable gene editing results than nuclease-based approaches but are far more difficult to program. The results of this study address an important hurdle in bringing site-specific recombinases closer to current nuclease technologies with regard to ease of implementation and accessibility for broader use.

The authors developed a deep learning algorithm that incorporates evaluation of the diversity of functional sequences across several Cre recombinase libraries, contributing a method and result that has not previously been reported. They convincingly validate their approach using a leave-one-out cross-validation, which confirms the utility of their method in predicting functional enzyme sequences on known targets. The in vitro analyses on predicted enzymes on novel targets is also convincing, but would benefit from 1) a larger set of enzymes, 2) more diversity in target sites, specifically with respect to the number of altered bases, 3) specificity analyses. With regard to the final point, the results presented in Fig 4B show activity, but a bottleneck that often emerges in recombinase engineering is that highly active enzymes may not be highly specific. Thus, further validation of the specificity of the four active recombinases would strengthen the results.

The findings of the study are potentially highly significant to the field of gene editing. Any means to increase the speed and ease of generating site-specific recombinases could expand and enhance the field of gene editing, which is currently dominated by nuclease-driven technologies that, although highly amenable to rapid deployment on virtually any new target, suffer from caveats related to the predictability of genomic outcomes. While the deep learning approach described here appears robust and valuable, at this point the results from the in vitro validation of the model still appear somewhat preliminary. In addition to the points noted above for suggested improvement of the manuscript, at only 40%, the actual yield of positive clones that perform as predicted is on the lower end of what can be considered significant, i.e., aiding a substantial increase in throughput for the process of generating enzymes capable of acting (specifically) on novel targets. Could further refinement of the algorithm or additional iteration improve these results?

The authors have clearly presented complex topics across multiple fields. The manuscript is well written and should be accessible to a broad audience and of particular interest to specialists in the site-specific recombinase sub-field of gene editing. The data analysis is clear and accurate – all conclusions are supported by results presented in the text and figures.

Reviewer #2:

Remarks to the Author:

The authors propose to use generative sequence models to design recombinases with specific DNA targets. The work has to be seen within the rapidly growing context of new computational approaches, which aim at exploiting large-scale sequence data and state-of-the-art machine learning techniques for protein design and optimisation, to complement more traditional approaches based on directed evolution (i.e. random mutations and selection) or (without relevance for the current paper) biophysical modelling. The paper is written in a very clear way, results are mostly convincing (but see my comments below), computationally designed sequences are experimentally tested – in my perception, the paper (and recombinase design) can become a reference for the field. Also thanks to exceptionally clear schematic figures like the teaser figure or

the first panels of most figures in the paper.

However, there are a number of comments, which I would ask the authors to consider carefully in their revision. My own expertise is in computational protein biology, I do not have prior knowledge on recombinases, so my report will mostly concentrate on the computational aspect and few (more naive) questions on the experiments.

Major comments:

1. References: The authors mostly cite computational literature on VAE, GAN and autoregressive models. For protein sequence modelling, also architectures like (restricted) Boltzmann machines (aka Potts models) and language models based on transformers have been used. Sequences generated from GAN, VAE and Potts models have been tested experimentally – as far as I know not with respect to fine functional specificities as discussed here, but I still think that this kind of literature should be covered (I did not check all references, but I see only Repecka et al. 2021 concerning the papers with experimental verification of generated sequences).
2. I agree with the authors that Conditional VAE offer a very elegant framework, well adapted to the computational problem of designing recombinases for given targets. One might consider other approaches, in particular autoregressive models are suitable to probabilistically model the recombinase sequence given the DNA target, or translation models (transformers) would “translate” targets to protein sequences and back. I don’t request authors to do tests with such models, but the introduction could be reworked.
3. Data: I would urge the authors to make the data easily available (the 89 libraries containing alignments of evolved recombinases and DNA targets). This is currently done only on request, and distribution on request is typically not very efficient. I would urge authors to deposit them in a repository, similarly to what is already done with the code and GitHub. I have the impression that these data can become an important benchmark dataset since they are quite unique. They are also essential for reproducing the results of the paper.
4. I wonder how difficult the recombinase design problem is at the end. It seems rather straight forward even if time consuming to evolve recombinases for specific DNA targets. Also the CVAE work with a very small two-dimensional latent space, this means that recombinases even for novel targets can be designed knowing only the target and two normally distributed random numbers. At this point I would intuitively expect that much simpler approaches like specific sequence motifs might give easier and more interpretable, but equally successful models - contrary to the authors speculation that “simple correlative analyses might not be sufficient”. There is a whole world between simple correlations and deep learning.
5. I think that the initial analysis of the evolved libraries is a bit superficial, and some numbers seem incoherent: (i) 2.2M sequences in 89 libraries correspond to an average of about 25k sequences per library. The numbers reported in Suppl. Fig. 1C look smaller, with an average of 10-15k per library. (ii) The authors use 1000 reads per library. How are these chosen? 1000 random reads, 1000 unique sequences? The answer could also clarify Supp. Fig. 2A. (iii) Fig. 1C could be improved, e.g. by using a sequence logo or a heat map for the mutational frequencies (i.e. for each of the 20 amino acids and each of the 343 sites, the frequency is displayed in a 2d array). This would be far more informative. (iv) It remains unclear if all libraries were evolved from the same initial Cre recombinase (authors say “Are-Based recombinases”). If they were different, this would totally change conclusions of these mutational spectra since mutations present in the initial sequence would differ from this acquired during directed evolution. (v) I am not at all convinced by the conclusion that all variable positions are under selection. According to basic evolutionary theory, depending on the library size, the mutation rate and selection strength, sequence variants might even reach high density by genetic drift, and even deleterious mutations are possible as passengers of beneficial driver mutations. This is true also for the newly found positions 5, 23, 57 and 166, and for the conclusions on selection on the N-terminal region previously found to be dispensable.

Specific comments:

6. Fig. 1D shows clustering of individual libraries. Even if t-SNE is not easily interpretable, this might indicate that there is a rather simple low-dimensional matching from recombinase sequence to its DNA target. It would be interesting if a similar picture appears when sequences are projected to the also 2d latent space of the CVAE. The picture would probably be a bit less clear

since the posterior latent distribution is required to be close to a 2d normal distribution by the loss function, but the analysis would be simple since the latent space is already 2d.

7. In the Methods section, the authors rederive the ELBO loss function. I have the impression that at this level of detail, it is accessible only for specialists, but these specialists know the derivation. A more qualitative description might be more adapted here, possibly with technical details in the supplement. On the other hand, the fact that from each library only 1000 recombinase sequence were selected, and the architectural parameters of the CVAE are given without motivation. Are results robust with respect to changes in the hyperparameters? Also, the one-hot encoding of the DNA target sequences as amino acids blows up the dimensionality of the data points. Why didn't the authors chose a more specific one-hot encoding in 4 dimensions for nucleotides?

8. It seem that the LOOCV tests perform systematically better than the novel designs. Can the authors comment on this observation?

9. It is a bit concerning that the generated PRLs are less diverse than the TRLs (Supp. Fig. 4A). Could the authors comment in more detail on this issue? How many sites are still variable in the PRLs?

10. If I understood well, recombinases should be specific, i.e. they should interact with their targets, but not with others. Most experiments (but part of Fig. 4B) address only the interaction with the desired target, but do not exclude promiscuous interactions. I would be happy to see the authors point of view on this question.

11. Not being an experimentalist, I have some problems interpreting part of the experimental results: (i) Activities are given in Fig. 3 and 4 via percentages. What do these percentages indicate? What would be a "good" number? (ii) Why are the designs for novelF-1 and novelF-2 not indicated as successful? They have bands in the desired region between 1.5 and 2 kb.

12. The construction of the novel target sites remains a bit unclear to me. Why is there only one variant for loxC-7 and loxD-7, but there are 5 for loxG-1?

Reviewer #3:

Remarks to the Author:

In this manuscript, Buchholz et al. described a machine learning algorithm termed RecGen (Recombinase Generator) for generating recombinases with new DNA target site selectivity. Briefly, 89 recombinase libraries were collected using PacBio HiFi sequencing technology, and these libraries and their respective target sites were used to train a conditional variational encoder (CVAE). The model's capability of predicting functional recombinases was first evaluated using leave-one-out cross-validation (LOOCV), by demonstrating the hamming distance between the predicted recombinase library (PRL) and target recombinase library (TRL) is generally smaller than that between closest recombinase libraries (CRL) and TRL. The authors also experimentally validated three target sites from LOOCV, and ten novel target sites, with a successful rate of 4/10. The manuscript was well written, and it possess novelty and significance for publication in Nature Communications if the authors can address the following comments.

Major comments:

1) The evaluation of RecGen using LOOCV is analyzed based on the assumption of small hamming distance meaning functional recombinase. But as the authors mentioned in line 138, some recombinases carried up to 75 amino acid substitutions. The current evidence does not fully support the robustness of RecGen. Experimental validations on libraries where d_H (CRL) is large should be included.

2) The authors use LOOCV testing to demonstrate CVAE is not "memorizing" the training set but infers the PRL sequences from training set. However, the median of d_H (CRL)- d_H (PRL) for all three experimental validations presented are less than 10. More evidence for libraries where the median of d_H (CRL)- d_H (PRL) is large should be presented to support that PRLs are inferred by CVAE.

3) Will the less frequent bases in certain positions of the half-site in the training dataset hinder the accuracy of RecGen? For example, half-sites with T at 3rd, T at 5th, G or C at 6th position. Experimental validations on these target sites may provide stronger support for the reliability and robustness of RecGen than randomly chosen target sites.

4) Is there a way to rationally rank the generated recombinases, instead of picking the most prevalent one?

Minor comments:

1) Neither the data necessary to train the model, nor the pre-trained models are provided in the GitHub page, making other researchers impossible to make new predictions using RecGen. (As of 04.07.2022)

2) When validating PRL for novel target sites, how did the PRLs perform compared with CRL on the 5 targets sites where CRLs had activity?

3) Figure 2B should also describe how the generated sequence as chosen based on prevalence.

4) Figure 2E, what is the logic to present these 17 particulars as a subset instead of presenting all libraries?

5) The math equations in line 500 and line 515 were not formatted properly.

Reviewer 1

The authors describe the development and preliminary testing of a deep learning approach to predict amino acid sequences of tyrosine recombinases that will have catalytic activity on novel target sites, based on previously existing libraries. Taking their approach from in silico predictions to in vitro analyses of a small set of predicted protein sequences, they observed recombination of novel target sequences in 4 of 10 of the enzyme/target pairs tested. This approach is substantially faster and more accurate than the alternative method of using the closest sequence as a starting point for evolution. Because tyrosine recombinases yield seamless genomic outcomes that don't rely on the stochastic outcomes of cellular repair machinery, they offer a means to achieve more predictable gene editing results than nuclease-based approaches but are far more difficult to program. The results of this study address an important hurdle in bringing site-specific recombinases closer to current nuclease technologies with regard to ease of implementation and accessibility for broader use.

Reply: We thank reviewer #1 for recognizing the significance of our work. We have addressed the points raised as outlined below.

The authors developed a deep learning algorithm that incorporates evaluation of the diversity of functional sequences across several Cre recombinase libraries, contributing a method and result that has not previously been reported. They convincingly validate their approach using a leave-one-out cross-validation, which confirms the utility of their method in predicting functional enzyme sequences on known targets. The in vitro analyses on predicted enzymes on novel targets is also convincing, but would benefit from 1) a larger set of enzymes, 2) more diversity in target sites, specifically with respect to the number of altered bases, 3) specificity analyses. With regard to the final point, the results presented in Fig 4B show activity, but a bottleneck that often emerges in recombinase engineering is that highly active enzymes may not be highly specific. Thus, further validation of the specificity of the four active recombinases would strengthen the results.

Reply: We thank reviewer 1 for the suggestions. We agree that testing a larger, more diverse set of enzymes would be beneficial. To this end, we have added additional experimental validations of one LOOCV prediction and four novel predictions, also addressing the diversity in the target sites. To fit the additional novel predictions, we have complemented Figure 4B to show all tested recombinases. With the additional data the figure became too large, which is why we cropped the gel pictures and included the full PCR gels in a new Supplementary figure (Supplementary Fig. 5).

Furthermore, we tested the novel recombinase predictions on their closest recombinase library target sites, which provides some insight into their specificity. However, we want to point out that the main goal of this work was to provide an algorithm that is capable of generating recombinases that have activity on a desired target site. We did not aim to produce highly specific recombinases. Also, during directed evolution approaches, the first goal is to achieve activity on a desired target site with specificity frequently found to be compromised (Buchholz and Stewart, 2001; Matsumura and Ellington, 2001). To gain specificity, methods for directed evolution and selection in mammalian cells have been described to solve this problem (Buchholz and Stewart, 2001; Karpinski et al., 2016). Nevertheless, it would be interesting to test whether RecGen could address this issue. However, libraries of recombinases selected for specificity would first have to be sequenced and the data analyzed by the algorithm. Possible adaptations to the code are likely necessary to achieve this.

We do not feel that this information is necessary for this manuscript and even believe that it would be distracting. But if the reviewer feels strongly about this point, we would be happy to add this information to the discussion.

The findings of the study are potentially highly significant to the field of gene editing. Any means to increase the speed and ease of generating site-specific recombinases could expand and enhance the field of gene editing, which is currently dominated by nuclease-driven technologies that, although highly amenable to rapid deployment on virtually any new target, suffer from caveats related to the predictability of genomic outcomes. While the deep learning approach described here appears robust and valuable, at this point the results from the in vitro validation of the model still appear somewhat preliminary. In addition to the points noted above for suggested improvement of the manuscript, at only 40%, the actual yield of positive clones that perform as predicted is on the lower end of what can be considered significant, i.e., aiding a substantial increase in throughput for the process of generating enzymes capable of acting (specifically) on novel targets. Could further refinement of the algorithm or additional iteration improve these results?

Reply: We want to thank the reviewer for recognizing the significance of our work. While the results of this work can be termed preliminary, we prefer to view RecGen as the first version of recombinase generator algorithms. We believe that further refinement of the algorithm and the training data set will lead to an improved algorithm that will reliably predict recombinases for desired target sites with little restrictions in terms of target site composition. We already point to potential improvement strategies in the discussion and we have enhanced the text to convey our opinion on the matter more clearly.

The authors have clearly presented complex topics across multiple fields. The manuscript is well written and should be accessible to a broad audience and of particular interest to specialists in the site-specific recombinase sub-field of gene editing. The data analysis is clear and accurate – all conclusions are supported by results presented in the text and figures.

Reply: We thank reviewer #1 for these final assessments.

Reviewer 2

The authors propose to use generative sequence models to design recombinases with specific DNA targets. The work has to be seen within the rapidly growing context of new computational approaches, which aim at exploiting large-scale sequence data and state-of-the-art machine learning techniques for protein design and optimisation, to complement more traditional approaches based on directed evolution (i.e. random mutations and selection) or (without relevance for the current paper) biophysical modelling. The paper is written in a very clear way, results are mostly convincing (but see my comments below), computationally designed sequences are experimentally tested – in my perception, the paper (and recombinase design) can become a reference for the field. Also thanks to exceptionally clear schematic figures like the teaser figure or the first panels of most figures in the paper.

However, there are a number of comments, which I would ask the authors to consider carefully in their revision. My own expertise is in computational protein biology, I do not have

prior knowledge on recombinases, so my report will mostly concentrate on the computational aspect and few (more naive) questions on the experiments.

Reply: We kindly thank the reviewer for complementing on the writing and the figures and on recognizing that this paper can become a reference for the field. Comments and suggestions from reviewer #2 were very valuable and we have addressed the points as outlined below.

Major comments:

1. References: The authors mostly cite computational literature on VAE, GAN and autoregressive models. For protein sequence modelling, also architectures like (restricted) Boltzmann machines (aka Potts models) and language models based on transformers have been used. Sequences generated from GAN, VAE and Potts models have been tested experimentally – as far as I know not with respect to fine functional specificities as discussed here, but I still think that this kind of literature should be covered (I did not check all references, but I see only Repecka et al. 2021 concerning the papers with experimental verification of generated sequences).

Reply: We indeed focused mostly on VAEs and GANs. At the time of conception of this project these were the major algorithms used for protein sequence generation. Direct Coupling Analysis (DCA) algorithms made from Potts models or Boltzmann machines have been used to predict the effect mutations have on the protein fitness, which can be leveraged to generate new viable protein sequences. However, due to their nature of direct residue comparisons, they are limited to lower order mutations and calculating combinations of mutations can get computationally very demanding. This is problematic when considering the task of predictions for target sites, where multiple bases are changed compared to known target sites, which affect multiple residues in the protein, in turn influencing each other. This is why we decided to go for a more holistic approach with generative deep learning models like VAEs or GANs. To keep the introduction to the point we wanted to focus on generative deep learning models without touching the, admittedly, very related field of DCA. However, we now included a short paragraph in the introduction that mentions DCA methods and why we were not using these kinds of models for our problem.

Furthermore, we mention several LSTM models in the introduction, which were inspired by language models, but we did not mention transformers. Transformers are very new in the field of protein sequence generation, but they are emerging to become an essential type of algorithm for this field. We adapted the introduction accordingly to better represent publications using this algorithm type.

Regarding models where experimental validation was performed, we never actually mention that any of these publications contain experimental validations, but all of the cited models that do conditional protein sequence generation either perform experimental validation or compare their predictions with previously tested sequences. However, since they are all predicting for very different proteins, we do not think it is necessary to go into their experimental validations. However, if the reviewer has a different opinion, we would be happy to expand the introduction.

2. I agree with the authors that Conditional VAE offer a very elegant framework, well adapted to the computational problem of designing recombinases for given targets. One might consider other approaches, in particular autoregressive models are suitable to probabilistically model the recombinase sequence given the DNA target, or translation models (transformers) would “translate” targets to protein sequences and back. I don’t request authors to do tests with such models, but the introduction could be reworked.

Reply: We thank reviewer#2 for this suggestion. The use of transformers for this project would indeed be very attractive. We have added additional text in the introduction to supplement this.

3. Data: I would urge the authors to make the data easily available (the 89 libraries containing alignments of evolved recombinases and DNA targets). This is currently done only on request, and distribution on request is typically not very efficient. I would urge authors to deposit them in a repository, similarly to what is already done with the code and GitHub. I have the impression that these data can become an important benchmark dataset since they are quite unique. They are also essential for reproducing the results of the paper.

Reply: We understand that the sequence data could become a useful resource for other scientists. However, it is currently not possible to deposit all sequences. A fraction of the sequence data stems from yet to be published projects. Public release of all the data would preclude protection of intellectual property for the ongoing projects. We hope that the reviewer understands our position on this matter. However, we have uploaded all recombinase sequence data of already published work on zenodo (<https://doi.org/10.5281/zenodo.7186203>). Further sequence data will be made available upon request with contractual limitations on how the data can be used and distributed so that the scientific community can evaluate and reproduce the results. Furthermore, we will make sequences publicly available as soon as the recombinase projects are published or protected.

4. I wonder how difficult the recombinase design problem is at the end. It seems rather straight forward even if time consuming to evolve recombinases for specific DNA targets. Also the CVAE work with a very small two-dimensional latent space, this means that recombinases even for novel targets can be designed knowing only the target and two normally distributed random numbers. At this point I would intuitively expect that much simpler approaches like specific sequence motifs might give easier and more interpretable, but equally successful models - contrary to the authors speculation that "simple correlative analyses might not be sufficient". There is a whole world between simple correlations and deep learning.

Reply: Unfortunately, the problem of developing new recombinases is so complex that it has not been solved in the last 20 years. We investigated the first order correlations between the recombinase sequences and the target sites, but could not make efficient use of the results for the design of novel recombinases. While we cannot exclude that different correlative analyses might be useful, we were unable to generate such models. Apparently, the multidimensionality of the possible combinations makes more interpretable approaches difficult to utilize. Since our goal was to achieve full sequence generation, we deemed a neural network model more convenient, even if they are difficult to interpret. We clarified the text sections in the introduction and in the results section "CVAE model architecture for recombinase prediction" to bring this point across better.

The latent space of the CVAE should only be representative of the sequence diversity observed in the recombinase library. The target sequence selectivity will be derived mostly from the target sequences that were concatenated with the latent space. The generation of the recombinases sequences itself of course was learned from the reconstruction loss to the input sequence. While it may seem like the decoder is only providing very little, it still has to learn the thousands of amino acid combinations of the recombinase sequences that are selective for a combination of 13 DNA bases.

5. I think that the initial analysis of the evolved libraries is a bit superficial, and some number seem incoherent: (i) 2.2M sequences in 89 libraries correspond to an average of about 25k sequences per library. The numbers reported in Suppl. Fig. 1C look smaller, with an average of 10-15k per library. (ii) The authors use 1000 reads per library. How are these chosen? 1000 random reads, 1000 unique sequences? The answer could also clarify Supp. Fig. 2A.

(iii) Fig. 1C could be improved, e.g. by using a sequence logo or a heat map for the mutational frequencies (i.e. for each of the 20 amino acids and each of the 343 sites, the frequency is displayed in a 2d array). This would be far more informative. (iv) It remains unclear if all libraries were evolved from the same initial Cre recombinase (authors say “Are-Based recombinases”). If they were different, this would totally change conclusions of these mutational spectra since mutations present in the initial sequence would differ from this acquired during directed evolution. (v) I am not at all convinced by the conclusion that all variable positions are under selection. According to basic evolutionary theory, depending on the library size, the mutation rate and selection strength, sequence variants might even reach high density by genetic drift, and even deleterious mutations are possible as passengers of beneficial driver mutations. This is true also for the newly found positions 5, 23, 57 and 166, and for the conclusions on selection on the N-terminal region previously found to be dispensable.

Reply: We thank reviewer #2 for these constructive points. (I) Based on this comment we checked our script again and found an error for counting the number of reads. We used the fasta format to store the sequence reads. To count the number of sequences, one can count the number of lines in the document and divide it by two. However, in the script used to count the lines of the fasta files, we forgot to divide by two. Therefore, the number of reads is exactly half of what we had reported in the original version of our manuscript. We have corrected the relevant sections. Thank you!

(II) The 1000 reads for training the algorithm were chosen randomly. The reads represented in supplementary figure 2A are exactly those chosen reads. We have added this information in the materials and methods section “CVAE model training and prediction”.

(III) We agree that to convey the sequence information, a better representation can be generated. We included a heatmap representation of the residues found at each given position. The amount of information included in this heatmap makes it difficult to get a good overview on the number of mutations that have been acquired, which is why we think Fig. 1C is still best suited as it is, but we are open to including the heatmap figure below this text (Fig. 1) as a supplementary image, if the reviewer feels strongly about it.

Fig. 1: Residues observed in all sequenced recombinases. Color indicates the total percentage of the residues per position over all 89 recombinase libraries.

(IV) Each evolution campaign for a new target site was initialized with a mix of recombinase libraries. Since multiple libraries were provided at the start of an evolution, a bias towards certain mutations was minimized as much as possible.

(V) We agree that the enriched residues are not necessarily beneficial for recombinase activity on the defined target site. However, the hot-spots we identified were found enriched in multiple evolutions, making it very likely that they are contributing to recombinase activity.

Additionally, previous work by Meinke et al. (Meinke et al., 2017) has shown that for the evolved Tre recombinase almost all of the acquired mutations were actually relevant for the recombination rate. Furthermore, work by Guillen-Pingarron et al. (Guillén-Pingarrón et al., 2022) has shown that the N-terminal region was specifically essential for the stabilization of the evolved recombinase. To properly reflect our opinion on this matter we improved the text in this regard (Discussion, page 17).

Specific comments:

6. Fig. 1D shows clustering of individual libraries. Even if t-SNE is not easily interpretable, this might indicate that there is a rather simple low-dimensional matching from recombinase sequence to its DNA target. It would be interesting if a similar picture appears when sequences are projected to the also 2d latent space of the CVAE. The picture would probably be a bit less clear since the posterior latent distribution is required to be close to a 2d normal distribution by the loss function, but the analysis would be simple since the latent space is already 2d.

Reply: When training a conventional VAE it is indeed possible to get decent representations of the recombinase library sequences, where the single libraries are separate clusters. Below is a visualization of the recombinase sequence training data with a two dimensional latent space (Fig. 2; same parameters as used for the CVAE model in the paper). The level of clustering of these points is comparable to the t-SNE dimensionality reduction.

Fig. 2: Visualization of a VAE latent space. The VAE was trained with a latent space size of 2, one dimension is shown per axis. Colors are according to the evolution project. Therefore multiple libraries are displayed in one color.

However, when we trained a CVAE on the recombinase sequences we used the target site as a condition. The target site could therefore be considered as an additional dimension to the latent space. As a result, the two dimensional latent space is not showing a clear clustering of the different libraries as can be seen in the CVAE latent space visualization of a

model trained with the same parameters as described in the publication (Fig. 3).

Fig. 3: Visualization of a CVAE latent space. The CVAE was trained with a latent space size of 2, one dimension is shown per axis. Colors are according to the evolution project. Therefore multiple libraries are displayed in one color.

7. In the Methods section, the authors rederive the ELBO loss function. I have the impression that at this level of detail, it is accessible only for specialists, but these specialists know the derivation. A more qualitative description might be more adapted here, possibly with technical details in the supplement. On the other hand, the fact that from each library only 1000 recombinase sequence were selected, and the architectural parameters of the CVAE are given without motivation. Are results robust with respect to changes in the hyperparameters? Also, the one-hot encoding of the DNA target sequences as amino acids blows up the dimensionality of the data points. Why didn't the authors chose a more specific one-hot encoding in 4 dimensions for nucleotides?

Reply: We agree that the ELBO rederivation is not necessary in the main text. We therefore moved it to the supplementary text as suggested. We used 1000 recombinase sequences to train the model, because it was close to the maximum of sequences we could use while still maintaining an even data distribution. We actually found that using 500 or less sequences already provided comparable results (data not shown). All other hyperparameters were determined by testing multiple setups followed by a grid search to determine the optimal combination of hyperparameters. Generally, changes in the hyperparameters resulted in minor changes in performance (data not shown). We chose the one-hot encoding for amino acids over the DNA one-hot encoding, because the model only needs to understand the changes in the protein, many of the DNA changes do not result in changes in the protein, which makes the information irrelevant for the model. Additionally, while the DNA encoding would reduce the input by 2960 values, a conversion to a non-sparse matrix, like amino acid property values or learned protein representations, would make more sense. We adapted the methods section to provide an overview of this information.

8. It seem that the LOOCV tests perform systematically better that the novel designs. Can the authors comment on this observation?

Reply: The LOOCV experimental validations were performed with predictions for target sites

where closely related recombinases were available in the training data set, with the closest recombinase library only being one or two base pairs different to the target site that was predicted for. The sequence information of these libraries should give the model an advantage in predicting recombinases for the LOOCV target sites compared to the novel target sites.

9. It is a bit concerning that the generated PRLs are less diverse than the TRLs (Supp. Fig. 4A). Could the authors comment in more detail on this issue? How many sites are still variable in the PRLs?

Reply: That the variance of the PRL is lower than the TRL is indeed interesting. VAEs are known to give “blurry” results. VAEs for image generation produce images that do not contain a lot of details. Learning features that are not represented very often do not seem to be cost effective for the models, which is why they tend to “generalize” the output. In this case we have a lot of features that are represented very rarely, directed evolution frequently produces mutations that have little effect on the functionality of the enzyme. The number of variable positions is generally above 200 or 300 in the evolved libraries, while when only considering mutations that occur in at least 5 % of the library we generally see less than 50 positions changed, which is much closer to the ~20 variable positions we observe in the predictions (LOOCV predictions). There are most likely other factors at play that influence this further, for example the model complexity and the amount of data available that is relevant for the predictions. We introduced some additional text to discuss this issue (page 18-19).

10. If I understood well, recombinases should be specific, i.e. they should interact with their targets, but not with others. Most experiments (but part of Fig. 4B) address only the interaction with the desired target, but do not exclude promiscuous interactions. I would be happy to see the authors point of view on this question.

Reply: We agree with the reviewer that this is an important point (see also point 1 of reviewer #1). Indeed, Matsumura and Ellington, as well as our own early work, showed that directed evolution for new functions frequently led to a loss in specificity or in other words an increase in promiscuity (Buchholz and Stewart, 2001; Matsumura and Ellington, 2001). Once recombinases are evolved for activity, they have to be further improved, either via additional directed evolution, or by rational design. The current version of RecGen only addresses activity on a specified target site. Hence, further directed evolution, or rational design on these recombinases will have to be performed to make these recombinases highly active and specific (Abi-Ghanem et al., 2012; Buchholz and Stewart, 2001; Hoersten et al., 2021; Karpinski et al., 2016; Lansing et al., 2022). We have added text to the discussion to address this point.

11. Not being an experimentalist, I have some problems interpreting part of the experimental results: (i) Activities are given in Fig. 3 and 4 via percentages. What do these percentages indicate? What would be a “good” number? (ii) Why are the designs for novelF-1 and novelF-2 not indicated as successful? They have bands in the desired region between 1.5 and 2 kb.

Reply: We thank reviewer #2 for these questions. (I) The percentages indicate how much of the plasmid was recombined in our test assay, it therefore gives the recombination rate of the recombinase at a given expression level. The higher the percentage the better. We have supplemented the text in the materials and methods section to clarify this point.

(II) PCR can lead to amplification of unwanted DNA products, which can be of the same size as the anticipated fragments. This would lead to a false positive identification of recombination. To ensure that all the products are correct, we confirmed all the potentially recombined DNA fragments with sequencing. The predictions for novelF-1 and novelF-2 did not provide a clear sequencing result and were therefore not included.

12. The construction of the novel target sites remains a bit unclear to me. Why is there only one variant for loxC-7 and loxD-7, but there are 5 for loxG-1?

Reply: We chose those target sites as a basis because they have been published previously. The uneven distribution of variants stems from generating predictions for loxG-1 target sites and then expanding to the other three target sites. The laboratory assembly of the target site vectors worked out better for loxF-1 target sites and when we had 10 in total we stopped making more. We now added additional novel target sites for loxC-7, loxD-7, and loxF-1 based target sites to even out the distribution a bit more.

Reviewer 3

In this manuscript, Buchholz et al. described a machine learning algorithm termed RecGen (Recombinase Generator) for generating recombinases with new DNA target site selectivity. Briefly, 89 recombinase libraries were collected using PacBio HiFi sequencing technology, and these libraries and their respective target sites were used to train a conditional variational encoder (CVAE). The model's capability of predicting functional recombinases was first evaluated using leave-one-out cross-validation (LOOCV), by demonstrating the hamming distance between the predicted recombinase library (PRL) and target recombinase library (TRL) is generally smaller than that between closest recombinase libraries (CRL) and TRL. The authors also experimentally validated three target sites from LOOCV, and ten novel target sites, with a successful rate of 4/10. The manuscript was well written, and it possess novelty and significance for publication in Nature Communications if the authors can address the following comments.

Reply: We thank reviewer 3 for complementing on the writing and for recognizing the significance of our work. We appreciate the comments and suggestions and have addressed the points made as follows:

Major comments:

1) The evaluation of RecGen using LOOCV is analyzed based on the assumption of small hamming distance meaning functional recombinase. But as the authors mentioned in line 138, some recombinases carried up to 75 amino acid substitutions. The current evidence does not fully support the robustness of RecGen. Experimental validations on libraries where d_H (CRL) is large should be included.

Reply: We believe this is a misunderstanding. The 75 amino acids changes refer to the number of amino acids that differ in an evolved recombinase to the original Cre sequence. Therefore, the recombinase with these 75 changes has activity on a very different target site. We always compare the predicted recombinases for a certain target site to recombinases where we know that they have activity on this target site. Nevertheless, we included the experimental validation of the LOOCV prediction for loxI-1, which has a large d_H (CRL). We also improved the text sections referring to the 75 amino acid changes, to clarify this point.

2) The authors use LOOCV testing to demonstrate CVAE is not “memorizing” the training set but infers the PRL sequences from training set. However, the median of d_H (CRL)- d_H (PRL) for all three experimental validations presented are less than 10. More evidence for libraries where the median of d_H (CRL)- d_H (PRL) is large should be presented to support that PRLs are inferred by CVAE.

Reply: We agree that testing another LOOCV sample with a larger d_H (CRL)- d_H (PRL) would be more convincing. As mentioned above, we now included the experimental validation for loxI-1, which has a larger d_H (CRL)- d_H (PRL).

3) Will the less frequent bases in certain positions of the half-site in the training dataset hinder the accuracy of RecGen? For example, half-sites with T at 3rd, T at 5th, G or C at 6th position. Experimental validations on these target sites may provide stronger support for the reliability and robustness of RecGen than randomly chosen target sites.

Reply: We thank reviewer #3 for this suggestion. To address this point, we have now tested a target site that contains changes in these positions and tested a prediction from RecGen on this site. The PCR assay uncovered the predicted band, indicating that RecGen is able to predict functional recombinases also on target sites where not many examples for these modifications were available from the training set. The results were incorporated into Fig. 4 in the revised version of the manuscript.

4) Is there a way to rationally rank the generated recombinases, instead of picking the most prevalent one?

Reply: We thank the reviewer for this interesting question. Currently, we are not aware of another method that would be more suitable. Our reason for choosing the most prevalent prediction is that if multiple models predict this recombinase with the highest frequency, then it must be the most “obvious” choice based on the data available, which is the best indication we have for a functional prediction. Ideally, we would like to test all the predicted recombinases as an artificial library, but we still have not established the best approach to generate a synthetic recombinase library. Nevertheless, we describe this possible improvement in the revised version of our manuscript (page 19).

Minor comments:

1) Neither the data necessary to train the model, nor the pre-trained models are provided in the GitHub page, making other researchers impossible to make new predictions using RecGen. (As of 04.07.2022)

Reply: We understand that the sequence data could become a useful resource for other scientists. However, it is currently not possible to deposit all sequences. A fraction of the sequence data stems from yet to be published projects. Public release of all the data would preclude protection of intellectual property for the ongoing projects. We hope that the reviewer understands our position on this matter. However, we have uploaded all recombinase sequence data of already published work on zenodo (<https://doi.org/10.5281/zenodo.7186203>). Further sequence data will be made available upon request with contractual limitations on how the data can be used and distributed so that the scientific community can evaluate and reproduce the results and make new predictions for designer-recombinases. Furthermore, we will make sequences publicly available as soon as the recombinase projects are published or protected.

2) When validating PRL for novel target sites, how did the PRLs perform compared with CRL on the 5 targets sites where CRLs had activity?

Reply: We thank reviewer #3 for this suggestion. We have included the testing of the novel recombinase predictions on their closest recombinase target sites (Supplementary Fig. 5B). Several of the functional predictions had no activity on their CRL target site (novelC-2, novelD-2, novelF-4), while some were active (novelF-5 and all of the novelG predictions). This indicates that even though we never required the model to make specific recombinases, it is still capable of predicting recombinases that are specific to some degree.

3) Figure 2B should also describe how the generated sequence as chosen based on prevalence.

Reply: We supplemented figure 2B with further explanations of how we proceeded after the predictions. Thank you.

4) Figure 2E, what is the logic to present these 17 particulars as a subset instead of presenting all libraries?

Reply: We showed a subset of the results, because showing all 89 LOOCVs would be too confusing and unclear. However, to address this point, we now show all 89 LOOCV results in Supplementary Fig. 4A.

5) The math equations in line 500 and line 515 were not formatted properly.

Reply: We checked all the equations and could not find any formatting errors. Is it possible that a different document viewer could show the equations incorrectly? We included the equations attached as an image, please specify if and where errors are present.

$$P(z|X) = \frac{P(X|z)P(z)}{P(X)}$$

$$P(X) = \int P(X|z)P(z)dz$$

$$Loss = -\frac{1}{n} \sum_{i=1}^n y_i \cdot \log \hat{y}_i + (1 - y_i) \cdot \log(1 - \hat{y}_i)$$

$$\log P(X) - D_{KL}[Q(z|X)||P(z|X)] = E[\log P(X|z)] - D_{KL}[Q(z|X)||P(z)]$$

$$\log P(X|c) - D_{KL}[Q(z|X, c)||P(z|X, c)] = E[\log P(X|z, c)] - D_{KL}[Q(z|X, c)||P(z|c)]$$

Fig. 3: Equations used in the submitted manuscript as image.

References

- Abi-Ghanem, J., Chusainow, J., Karimova, M., Spiegel, C., Hofmann-Sieber, H., Hauber, J., Buchholz, F., Pisabarro, M.T., 2012. Engineering of a target site-specific recombinase by a combined evolution- and structure-guided approach. *Nucleic Acids Res.* 41, 2394–2403. <https://doi.org/10.1093/nar/gks1308>
- Buchholz, F., Stewart, A.F., 2001. Alteration of Cre recombinase site specificity by substrate-linked protein evolution. *Nat Biotechnol* 19, 1047–1052.
- Hoersten, J., Ruiz-Gómez, G., Lansing, F., Rojo-Romanos, T., Schmitt, L.T., Sonntag, J., Pisabarro, M.T., Buchholz, F., 2021. Pairing of single mutations yields obligate Cre-type site-specific recombinases. *Nucleic Acids Res.* <https://doi.org/10.1093/nar/gkab1240>
- Karpinski, J., Hauber, I., Chemnitz, J., Schäfer, C., Paszkowski-Rogacz, M., Chakraborty, D., Beschorner, N., Hofmann-Sieber, H., Lange, U.C., Grundhoff, A., Hackmann, K., Schrock, E., Abi-Ghanem, J., Pisabarro, M.T., Surendranath, V., Schambach, A., Lindner, C., Lunzen, J. van, Hauber, J., Buchholz, F., 2016. Directed evolution of a recombinase that excises the provirus of most HIV-1 primary isolates with high specificity. *Nat. Biotechnol.* 34, 401–409. <https://doi.org/10.1038/nbt.3467>
- Lansing, F., Mukhametzhanova, L., Rojo-Romanos, T., Iwasawa, K., Kimura, M., Paszkowski-Rogacz, M., Karpinski, J., Grass, T., Sonntag, J., Schneider, P.M., Günes, C., Hoersten, J., Schmitt, L.T., Rodriguez-Muela, N., Knöfler, R., Takebe, T., Buchholz, F., 2022. Correction of a Factor VIII genomic inversion with designer-recombinases. *Nat. Commun.* 13. <https://doi.org/10.1038/s41467-022-28080-7>
- Matsumura, I., Ellington, A.D., 2001. In vitro evolution of beta-glucuronidase into a beta-galactosidase proceeds through non-specific intermediates. *J. Mol. Biol.* 305, 331–339. <https://doi.org/10.1006/jmbi.2000.4259>

Reviewers' Comments:

Reviewer #1:

Remarks to the Author:

The authors have thoroughly addressed comments and concerns from all reviewers. I have no further comments or suggested revisions.

Reviewer #2:

Remarks to the Author:

The authors have carefully revised their manuscript following the reviewers' indications (or explaining why they rather preferred the initial presentation). They have even extended the cases used for experimental validation.

In my eyes, there are only two points, the first one emerging in the revision, the second one remaining from the original manuscript:

(1) I think the comments about DCA-like models are not correct. These models are generative and explicitly score arbitrary sequences, including thus arbitrary numbers of mutations with respect to some reference. To give some examples McGhee et al. Nature Comm 2021 have compared the generative properties of DCA and VAE, Russ et al. Science 2020 have generated diversified enzyme sequences and tested them experimentally, and de la Paz et al. PNAS 2020 have used DCA models to describe protein evolution over many mutations. So I guess they could be a similarly valid choice as the CVAE used here.

(2) I fully understand the point the authors make about data availability, but it makes the results not reproducible for the readers, and limits the possibility to test alternative approaches on the same data. How much could actually be achieved with the data provided? I think it should also be checked with the editorial office, if I remember well Nat Comm has rather well defined rules about data availability.

Once point (1) being corrected and point (2) addressed, I recommend publication of the manuscript.

Reviewer #3:

Remarks to the Author:

The authors have fully addressed my critiques. The revised manuscript can now be accepted for publication.

REVIEWERS' COMMENTS

Reviewer #1

The authors have thoroughly addressed comments and concerns from all reviewers. I have no further comments or suggested revisions.

Reply: We are pleased to read that all points have been addressed. We thank Reviewer #1 for the valuable input.

Reviewer #2

The authors have carefully revised their manuscript following the reviewers' indications (or explaining why they rather preferred the initial presentation). They have even extended the cases used for experimental validation.

In my eyes, there are only two points, the first one emerging in the revision, the second one remaining from the original manuscript:

(1) I think the comments about DCA-like models are not correct. These models are generative and explicitly score arbitrary sequences, including thus arbitrary numbers of mutations with respect to some reference. To give some examples McGhee et al. Nature Comm 2021 have compared the generative properties of DCA and VAE, Russ et al. Science 2020 have generated diversified enzyme sequences and tested them experimentally, and de la Paz et al. PNAS 2020 have used DCA models to describe protein evolution over many mutations. So I guess they could be a similarly valid choice as the CVAE used here.

Reply: We thank Reviewer #2 for the input regarding the DCA-like models. We modified the wording of the DCA-related paragraph in line with the comments from reviewer #2.

(2) I fully understand the point the authors make about data availability, but it makes the results not reproducible for the readers, and limits the possibility to test alternative approaches on the same data. How much could actually be achieved with the data provided? I think it should also be checked with the editorial office, if I remember well Nat Comm has rather well defined rules about data availability.

Once point (1) being corrected and point (2) addressed, I recommend publication of the manuscript.

Reply: We understand the concerns regarding reproducibility and further research on the data. We have redefined the data availability statement according to the Nature Communications rules.

Reviewer #3

The authors have fully addressed my critiques. The revised manuscript can now be accepted for publication.

Reply: We are pleased to read that all points have been addressed. We thank Reviewer #3 for the valuable input.